# Refractory Pediatric Psoriasis and Atopic Dermatitis: The Importance of Therapeutical Adherence and Biological Management

**DOI:** 10.3390/biomedicines9080958

**Published:** 2021-08-04

**Authors:** Katherine A. Kelly, Adaora Ewulu, Veronica K. Emmerich, Courtney E. Heron, Steven R. Feldman

**Affiliations:** 1Center for Dermatology Research, Department of Dermatology, Wake Forest School of Medicine, Winston-Salem, NC 27103, USA; are45@georgetown.edu (A.E.); vemmeric@wakehealth.edu (V.K.E.); courtneyheron1@gmail.com (C.E.H.); sfeldman@wakehealth.edu (S.R.F.); 2Department of Pathology, Wake Forest School of Medicine, Winston-Salem, NC 27103, USA; 3Department of Social Sciences & Health Policy, Wake Forest School of Medicine, Winston-Salem, NC 27103, USA; 4Department of Dermatology, University of Southern Denmark, 5230 Odense, Denmark

**Keywords:** resistance, management, adherence, anti-drug antibodies, biologics, psoriasis, atopic dermatitis

## Abstract

The rates of refractory pediatric psoriasis and atopic dermatitis (AD) have steadily risen over the last few decades, demanding newer and more effective therapies. This review aims to explore the reasons for resistant disease, as well as its management; this includes the indications for, efficacy of, and safety of current therapies for refractory pediatric dermatologic disease. A PubMed search for key phrases was performed. Poor medication adherence is the most common cause of resistant disease and may be managed with techniques such as simplified treatment regimens, more follow-ups and educational workshops, as well as framing and tailoring. Once problems with adherence are ruled out, escalating treatment to stronger biologic therapy may be indicated. Development of anti-drug antibodies (ADAs) can cause patients’ disease to be refractory in the presence of potent biologics, which may be addressed with regular medication use or concomitant methotrexate. If patients with AD fail to respond to biologic therapy, a biopsy to rule out mycosis fungoides, or patch testing to rule out allergic contact dermatitis, may be indicated. A limitation of this study is the absence of more techniques for the management of poor medication adherence. Managing medication adherence, escalating treatment when appropriate, and addressing possible anti-drug antibodies will help assure control and relief for patients with resistant disease.

## 1. Introduction

Atopic dermatitis (AD) and psoriasis are two chronic skin conditions that are commonly seen by dermatologists [1]. The prevalence of psoriasis is 2–4% in North America. It is associated with poor quality of life in patients, as well as significant psychological and economic stressors. AD has a lifetime prevalence of 15–30% in children, which persists to 8% in adulthood. AD is also associated with poor quality of life, and often affects sleep, self-esteem, and school and professional performances. Psoriasis and AD also increase the risk of other chronic conditions, such as diabetes and cardiovascular disease. Topical treatments are often the mainstay treatments for these conditions, but systemic therapies may be used for more severe cases.

Treatments for AD are often multifaceted and can include emollients, topical corticosteroids, calcineurin inhibitors, trigger avoidance, and phototherapy, as well as oral anti-inflammatory agents and biologic agents in severe cases [2]. Psoriasis responds to some similar treatments—such as topical corticosteroids, calcineurin inhibitors, phototherapy, and biologic agents—and may also be managed with vitamin D3 analogues, synthetic retinoids such as acitretin, and methotrexate [3].

Topical corticosteroids are commonly used to treat both AD and psoriasis, but many patients develop “resistance” over time [4]. Topical steroids are usually associated with vitamin D derivatives in the management of psoriasis [5]. In theory, resistance can be caused by a downregulation of target receptors due to chronic drug exposure; however, downregulation of target receptors has not been demonstrated in a clinical setting [4]. Reduced medication adherence may play a greater role in the development of resistant disease in this patient population as compared to the downregulation of target receptors [4]. This manuscript will explore the causes of resistant AD and psoriasis, as well as management options.

## 2. Materials and Methods

A PubMed and SCOPUS search included the keywords refractory, psoriasis, atopic dermatitis, eczema, pediatric, child, biologics, treatment, methotrexate, recalcitrant, dupilumab, mycosis fungoides misdiagnosis, contact dermatitis, adherence in atopic dermatitis, adherence in psoriasis, biologics in atopic dermatitis and psoriasis, management of atopic dermatitis, management of psoriasis, anchoring and adherence in atopic dermatitis, anchoring and medication adherence, framing and medication adherence, biologics and pediatric psoriasis, dupilumab and atopic dermatitis, anti-drug antibodies and atopic dermatitis, anti-drug antibodies and psoriasis, methotrexate use with biologics in atopic dermatitis and psoriasis, patch testing in atopic dermatitis management, resistant disease and medication adherence in atopic dermatitis and psoriasis, adherence, biologics, steroid phobia, complementary and alternative medicine atopic dermatitis, and complementary and alternative medicine psoriasis.

## 3. Results

There are multiple potential causes of resistance, some that may require escalation of treatment or further testing. Medication adherence, the need for stronger drugs, the development of anti-drug antibodies, and indications for additional diagnostic workup will be discussed.

### 3.1. Poor Adherence

#### 3.1.1. Poor Adherence Is the Most Common Cause of Resistant Disease

Adherence to treatment can be categorized into primary and secondary adherence. Primary adherence refers to getting and starting the treatment. Increased cost and treatment regimen complexity are associated with lower prescription fill rates. In a study of 143 patients with acne, patients prescribed a topical treatment consisting of one medication filled their prescription 91% of the time [6]. In contrast, patients prescribed two medications filled their prescriptions only 60% of the time. In a similar study, 322 patients with various dermatologic conditions were given a new prescription; of these, 30.7% did not redeem at least one of their medications and 29.2% did not redeem any of them [7]. Psoriasis patients had the lowest level of primary adherence with 50% of prescriptions unredeemed [8]. Those who were male, elderly, on systemic medications, and with short-term diseases, such as infections or mycoses, had the greatest level of primary adherence [8]. 

Secondary adherence refers to what happens after a patient initiates treatment; it includes how closely the patient uses the medication as prescribed and whether they discontinue treatment at the appropriate time. Secondary adherence can be assessed by looking at refill rates, self-reported treatment diaries, or, more objectively, through use of electronic monitoring devices such as Medication Event Monitoring System (MEMS) caps, which are specialized packages that electronically record the time of dosing events.

#### 3.1.2. Adherence to Topical Treatment Is Inadequate

Adherence to topical therapy is particularly poor—in a study in which participants were instructed to use a topical medication twice daily for 12 weeks, MEMS data for the 3rd week revealed that the bottle was opened twice daily on only 35% of days [8]. Interventions that require more patient participation, such as the self-reporting of medication experience, may increase secondary adherence in the long term. In a study evaluating topical fluocinonide adherence over 12 months among patients with mild–moderate psoriasis, those who reported their perception of symptom improvement over the internet each week demonstrated greater adherence (50%) compared to those in the standard-of-care group (35%) [9]. This increase in adherence was also associated with greater improvements in Psoriasis Area and Severity Index (PASI) scores in the intervention group at one, three, and twelve months [10].

A barrier to adherence among pediatric patients prescribed topical corticosteroids is “steroid phobia”. A majority of parents and caregivers of children with AD express aversion to using topical corticosteroids due to fear of adverse effects [10]. In a real-world use study of 26 pediatric patients with AD, overall mean adherence to topical triamcinolone was a mere 32% over the course of eight weeks, with a precipitous drop in adherence over the few days following the initial office visit.

#### 3.1.3. Adherence to Biologics Is Poor

Adherence to biologic treatments also decreases over time. In a systemic analysis of adherence to biologic therapies for psoriasis, adherence was defined as the medication possession ratio being over 80%, or as the medication being used as prescribed for more than 80% of days. By this metric, only 49%, 41%, and 29% of patients prescribed infliximab, adalimumab, and etanercept, respectively, were adherent [11]. In a one-year open-label trial, the use of extended nurse education was evaluated for its effect on adherence to adalimumab in psoriasis patients [12]. There was great variability in the number of doses taken and the days between doses in both patients who received additional nurse education as well as those who received standard physician education materials alone [13].

There are many barriers to biologic use. Biologics are expensive—adalimumab is estimated to cost $72,000 United States dollars (USD) annually—and patients frequently have difficulty with insurance coverage. Some patients may attempt to conserve medication by increasing dosage intervals or by only using their medication when experiencing a disease flare [13]. Other barriers to biologic use include fear of adverse effects and pain during injection. For pediatric patients especially, the fear of needles must be overcome.

#### 3.1.4. Improving Adherence in “Resistant” Patients Overcomes Resistance

Tachyphylaxis is defined as a rapid loss of response due to the downregulation of receptors; the term has been used in dermatology to refer to the slow loss of effectiveness of topical corticosteroids when used for chronic inflammatory skin diseases [14]. The decreased effectiveness in the long-term may be due more to poor use of the medication over time, rather than to the downregulation of steroid receptors [15]. In a case report of a patient with severe “coral reef” psoriasis who had failed etanercept and oral prednisone therapies, dramatic improvement was achieved with 0.05% clobetasol propionate spray when the patient was instructed to use the spray twice daily and return for a follow-up visit in three days [14]. No keratolytic agent was required. In a small study of patients with psoriasis who were “resistant” to topical corticosteroids, a good response to topical corticosteroids occurred under conditions designed to promote good adherence. Twelve “topical steroid resistant” psoriasis patients treated with 0.25% desoximetasone spray received either twice-daily phone call reminders or no reminders over two weeks [16]. Patients in the reminded group demonstrated improvement in their Pruritus Visual Analog Scale (VAS), PASI, Total Lesion Severity Score (TLSS), and Investigator Global Assessment (IGA) scores [4]. Similar studies evaluating the effect of twice daily phone reminders on symptom severity were completed with “topical corticosteroid resistant” AD patients [4,17]. Those who received reminders showed significantly greater mean improvement in their Eczema Area and Severity Index (EASI), TLSS, and IGA scores by the third visit, as compared to those who did not receive reminders [4,17]. These results suggest that “resistant” disease in AD and psoriasis is largely the result of poor adherence.

#### 3.1.5. Approaches for Improving Adherence

##### Improving the Foundation for Good Adherence: Trust and Accountability

A strong doctor–patient relationship is one of the best predictors of adherence to AD treatment [17]. Effective verbal and nonverbal communication, active listening, and collaboration all contribute to an optimal doctor–patient relationship and improve patient satisfaction [2]. Taking the time to educate patients about their disease and medications increases their understanding of their condition and the reasons to remain adherent to treatment.

A follow-up visit, or other planned contact with the patient, shortly after initiating a new treatment may foster a sense of accountability that encourages patients to fill the prescription right away and to use the medication well. In clinical trials, there are generally follow-up visits at weeks 1, 2, 4, 6, 8, etc., that encourage better adherence. In clinical practice, patients are often told to use the medication and return in 8–12 weeks, and their adherence suffers. A single early-return visit can be a powerful motivator to start on treatment and use it well. When patients see that the medication works, they may be encouraged to use it as needed in the long run.

Similar to students who increase their practice time before their piano lessons, “white coat adherence” describes the phenomenon of patients increasing their adherence to a medication before follow-up appointments with their physician [18]. The mechanism of this phenomenon is possibly due to an increased sense of accountability among patients, or perhaps a decreased perception of treatment burden since they take the medication for a shorter amount of time before their next visit [19]. White coat adherence commonly occurs in patients with chronic dermatologic conditions and formed the basis of a study by Sagransky et al., which focused on the effect of scheduling follow-up visits at one-week intervals shortly after starting therapy for AD [20]. Those patients who had more follow-up visits showed greater adherence (88% at week one decreased to 50% at week four) as compared to the control group (83% at week one decreased to 43% at week four); however, the differences were not statistically significant [20]. Physicians may tend to schedule follow-up visits at 6–8 weeks or longer, in part perhaps because clinical trials often evaluate their primary results at this time period. However, many dermatologic treatments, if used correctly, should produce some improvement within a week or less [20]. It may benefit patients with chronic dermatologic conditions to have a follow-up appointment 1–2 weeks after new treatments are prescribed, in order to promote better medication adherence [20]. 

##### Standard Approaches to Improve Adherence

The recommended treatment for AD often includes complex lifestyle changes and multiple medications, which can make it difficult for patients to be adherent. AD action plans, which involve written treatment instructions, are useful for parents of children with eczema. These written instructions were cited by 68% of parents as a factor in the improvement of their child’s eczema [17].

Educational workshops, which provide parents with more detailed instructions through written materials and demonstrations of medication application, are an emerging technique to increase adherence in this population. Longer and more frequent workshops are associated with increased adherence. An educational workshop lasting two hours was more effective than those lasting 15–30 min [17]. In a study of 51 pediatric AD patients whose parents received spoken and written education, as well as a demonstration of medication application from a dermatology nurse, at every follow-up visit for one year, 77% of these patients used their topical medications at the correct quantity and 89% demonstrated a decrease in mean disease severity scores [17].

Caregivers of children with AD were surveyed regarding medication adherence; among those who did not administer their medication as directed, or stopped using it, “concern for side effects” was the most common reason for their behavior [21]. When asked about what could convince them to use the medication as prescribed, a “clearer indication of the effectiveness” was the most common response (55%), as well as “access to research or evidence about the benefit and side effect profile” (14%) [21]. Topical corticosteroids are often a mainstay of treatment for AD; however, over 80% of parents of children with AD demonstrate corticosteroid phobia, or a fear of potential side effects of frequent steroid application [2]. Skin atrophy, immune suppression, and growth delay are common concerns among parents [2]. Corticosteroid phobia is associated with an increased need for reassurance in patients with AD as well as their caregivers [17].Creating a strong, trusting relationship between the physician and the patient and addressing the concerns around side effects may help to increase adherence in this population [17].

AD patients and their caregivers may pursue complementary and alternative medicines (CAMs) if they are concerned with possible side effects or in other ways dissatisfied with their treatment regimen [22]. Over 40% of pediatric patients with AD report the use of alternative medicine approaches [22]. However, among patients who used CAMs for skin-related conditions, 42.3% did not report this to their physician, either because their physician never inquired about their use of CAMs or they did not believe that the physician needed to know [23]. There are a number of CAMs available for dermatologic conditions, but herbal therapies are the most common [24]. Although 82.6% of respondents self-reported improvement with the use of CAMs, particularly herbal therapies, studies evaluating the effects of CAMs on psoriasis patients exhibited worse severity in those who tried herbal remedies, vitamin therapy, and dietary changes [24,25]. Other CAM approaches are better validated in the literature, particularly the use of probiotics for the treatment of pediatric AD [22]. The most effective treatment is with *L. plantarum* and *L. fermentum* in children 12 months of age and older, likely from assisting the development of the immune system with microbial stimulation [22]. The efficacy of this treatment was demonstrated with a clinically significant improvement in the SCORing AD (SCORAD) index of 8.7 points, which equates to a one-point improvement on the global severity scale Discussing CAM use with patients provides physicians with an opportunity to educate patients with evidence-based data regarding the types of CAM that could augment their treatment and to prevent them from becoming nonadherent to their existing regimen.

##### Advanced Adherence Interventions

Patient’s perceptions of treatment regimens may contribute to differences in adherence, which forms the basis of studies on tailoring techniques [26]. Personalization of treatment plans communicates to patients that the instructions of their treatment regimen are designed specifically for them (Figure 1) [26]. Placebo tailoring describes giving patients an artificially customized treatment plan; this plan appears to be tailored to the specific patient when it is not [26]. Placebo tailoring increases patient satisfaction with and willingness to follow treatment plans [26]. 

Perceptions of treatment burden may also affect patient and caregiver adherence if they believe that the treatments are too complex and time-consuming to perform consistently over time. In a study using anchoring to modify patients’ perception of burden, patients with psoriasis were more willing to use an injectable medication once a month if they were previously asked about using an injectable medication once a day [27]. Similar studies among caregivers of patients with AD have assessed whether anchoring to a more extensive treatment regimen improved caregivers’ adherence to their actual treatment regimens [27]. One of these studies compared the willingness of caregivers of AD patients to treat their child’s AD with a once-daily treatment between those previously asked to treat their child with a four times daily treatment and those who were only asked about the once daily treatment [26]. Among the caregivers previously asked about the four times daily medication, there was an increase in their willingness to treat with a once daily topical corticosteroid; however, this difference was not statistically significant [27]. 

Educating patients on side effects before starting new treatments increases adherence, and framing negative side effects as indicators for positive outcomes and evidence of treatment efficacy further increases this effect [28]. In a recent survey study, 1039 AD patients were randomly assigned to three hypothetical scenarios in which they were given a topical medication and experienced one of three side effects [28]. They experienced either a painful sensation, a non-painful sensation, or no sensation. Each group reported on a 9-point Likert-type scale how willing they would be to continue the use of the medication. Their willingness to continue to use the medication increased when they were educated by their physician about the possible side effect (*p* < 0.001; d = 0.46) [28]. Their willingness was even greater when the unpleasant sensation was framed as an indicator of efficacy (*p* < 0.001, d = 1.32) [28]. The hurdle of side effects can be made an advantage through framing [28]. 

### 3.2. When Stronger Drugs Are Needed

Some patients do not respond even when they are using the treatment properly. Dupilumab revolutionized the treatment of AD (Table 1). Multiple biologics are available for the treatment of pediatric psoriasis that provide specific inhibition of immune-mediated pathways involving tumor necrosis factor (TNF), IL-17, and IL-23 (Table 2) [29]. A recent meta-analysis compared the short- and long-term efficacy of multiple available biologics by measuring the PASI score at 10–16 weeks and 44–60 weeks [29]. After both time points, risankizumab-rzaa, brodalumab, ixekizumab and guselkumab had the highest PASI 90 scores [29]. 

### 3.3. Addressing Anti-Drug Antibodies

Development of anti-drug antibodies (ADAs) can cause patients’ disease to be refractory to otherwise potent biologic treatments. ADAs are associated with an increased risk of infusion reactions, quicker drug clearance, and a decreased response to the drugs. Preventive strategies include taking medication regularly with maintenance of a stable therapeutic trough drug concentration and coadministration of an immunosuppressive such as methotrexate [42]. Although biologics should be administered as continuous therapy with regular fixed time intervals, actual use in patients may vary. Given external factors such as patient adherence, lifestyle, and financial constraints, studies examining therapeutic drug monitoring in real-world settings are important in shedding light on drug survival [43]. Involvement of a patient in all aspects of treatment selection ensures high retention rates and stable therapeutic trough drug concentrations [43]. Measuring medication trough levels in patients who are not responding well to biologics could potentially help to determine whether to switch to a different biologic drug or to alter the dose of the drug being used. However, this approach is not commonly taken in psoriasis or AD treatment [44]. 

Concomitant immunomodulators, such as methotrexate and azathioprine, can reduce the immunogenicity of therapeutic antibodies in many inflammatory conditions [44]. In a cohort study of 121 patients with rheumatoid arthritis (RA) treated with adalimumab, anti-adalimumab antibodies were detected in 17% of patients during 28 weeks of treatment [45]. Treatment non-responders had more ADAs in their system as compared to good responders [45]. Additionally, patients with ADAs had lower serum adalimumab drug concentrations by the end of the 28th week [45]. Concomitant methotrexate use was higher in the group without antibodies as compared to the group with anti-adalimumab antibodies (84% vs. 52%) [45]. 

In a prospective study of patients with Crohn’s disease, the concomitant use of immunosuppressive therapy in addition to infliximab was compared with infliximab monotherapy in 174 patients [46]. In one study arm, 65 patients used azathioprine (2 to 2.5 mg/kg); in a second arm, 50 patients used 15 mg intramuscular (IM) or subcutaneous (SC) methotrexate weekly; and in a third arm, 59 patients received infliximab monotherapy [46]. With concomitant use of either azathioprine or methotrexate, incidence of ADAs was lower as compared with patients using infliximab therapy alone [46]. 

Methotrexate use with biologics for the prevention of ADAs in patients with psoriasis has not been extensively studied. The addition of immunosuppressants to a treatment regimen carries further risk of adverse reactions from a reduced immune response, which should be weighed against the benefit of preventing ADAs [47]. 

### 3.4. Resistant Disease Can Be Due to the Wrong Diagnosis

If AD does not respond to dupilumab, biopsies to rule out mycosis fungoides, and allergy testing to rule out contact dermatitis, should be considered. Mycosis fungoides is the most common form of cutaneous T cell lymphoma. It is a progressive condition characterized by fine, wrinkly, cigarette-paper scales forming scaly, erythematous plaques [48]. It can be difficult to clinically differentiate between mycosis fungoides and AD [48,49].

A study of 22 patients who were diagnosed with cutaneous T-cell lymphoma (CTCL) after failing to respond to TNF-a inhibitors found that ¾ of patients were being treated for what they believed were inflammatory skin conditions (mainly psoriasis and AD). Of these patients, 91% had biopsies consistent with mycosis fungoides, suggesting that initial clinical impressions of this disease may appear similar to inflammatory skin disorders [49]. In a recent case report, two patients with diffuse, pruritic, cutaneous eruptions were initially diagnosed with AD [50]. They were refractory to dupilumab after five months of treatment and subsequently biopsied; the pathologic findings were consistent with mycosis fungoides [49]. Mycosis fungoides may initially present as patches that change to plaques over time [49]. These initial lesions can look similar to other inflammatory cutaneous conditions, such as AD.

Clinicians may also perform allergy patch testing to try to rule out a common differential diagnosis for AD—allergic contact dermatitis [50]. Contact dermatitis can present as erythema, blister formation, weeping, and crusting. Patch testing may be considered once patients with AD fail topical treatments, but before beginning immunosuppressive therapy [50]. In particular, patch testing is indicated in adolescent or adult-onset AD, or in patients with an atypical lesion distribution. Patch testing in children or adults with AD that worsens with treatment or returns quickly upon stopping therapy should be considered, due to the possibility that they could have acquired allergic contact dermatitis from use of the topical medication [50]. 

## 4. Conclusions

AD and psoriasis are chronic dermatologic conditions that have a great impact on the lives of patients and their caregivers. The use of numerous medications, separate administration timings, and application of multiple topicals all contribute to treatment burden, which presents a hurdle to adherence—especially over the long run [2]. Addressing caregiver perceptions of treatment burden and complexity may be beneficial for medication adherence.

The most useful medications may be the ones that patients and their caregivers will take. If patients are not showing signs of symptom improvement, ensure that they are taking their medications as prescribed before escalating their treatment. Administering a topical steroid twice daily may seem difficult, especially if the patient is thinking about how difficult it will be to take the medication for an extended period of time. Therefore, it may be useful for physicians to schedule some type of follow-up within one week of starting a new medication, leaving patients thinking that the burden of treatment is only for a week and not indefinitely. Early follow-up is a strong motivator for better adherence, resulting in better treatment outcomes which may in turn motivate patients to use the medication as needed in the future [51].

Once adherence is addressed, escalating treatment beyond first-line topical therapies may be necessary. Moderate to severe AD or psoriasis may be treated in a stepwise approach with systemic therapies, phototherapy, or biologics. Stronger biologic drugs indicated for treating pediatric psoriasis include adalimumab, secukinumab, etanercept, ustekinumab, and ixekizumab. These drugs provide specific inhibition of immune-mediated pathways involving TNF, IL-17, and IL-23 [52]. Treatment options for AD include dupilumab, which inhibits IL-4 and IL-13. Biologics are effective therapeutic options, and they are generally very safe. The development ADAs can prevent a patient’s disease from improving in response to treatment. Preventative strategies include taking medication as regularly as prescribed and concomitant use of methotrexate.

If a patient presents with presumed AD that fails to respond to dupilumab, then a biopsy may be useful to rule out another disease such as mycosis fungoides. When treatment is not effective, allergy testing may be helpful in ruling out allergic contact dermatitis. Addressing medication adherence in conjunction with the proper use of available treatments may help assure effective symptom control and relief from resistant disease.

## Figures and Tables

**Figure 1 biomedicines-09-00958-f001:**
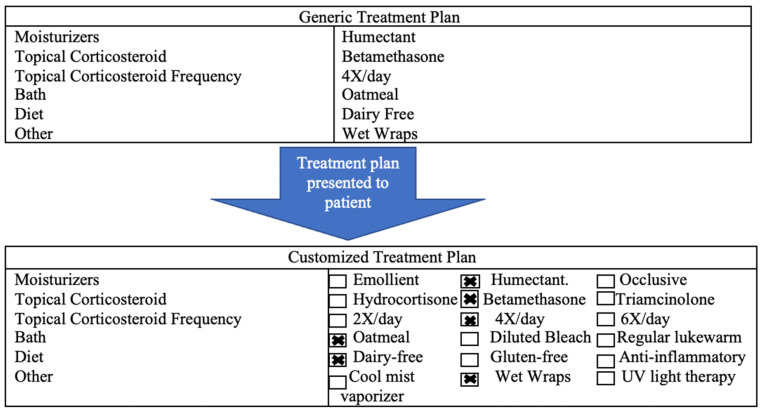
Generic treatment plan compared to placebo-tailored (individualized) treatment plan.

**Table 1 biomedicines-09-00958-t001:** Biologics indicated for the treatment of atopic dermatitis in children.

Drug Name	Biologic Agent Structure	Pediatric Indication	Study Type	Age of Participants	Length of Time	Methods	Results of Clinical Trials
IL-4 receptor a chain inhibitor							
Dupilumab	Fully human monoclonal IgG4 antibody that blocks IL4/IL13 signaling [30]	Patients age 6–11 with moderate-to-severe atopic dermatitis [31]	Phase III R, PC, DB	Children aged 6–11 years with severe AD inadequately controlled with topical therapies [31]	16 weeks	Patients were randomized 1:1:1 to 300 mg of dupilumab every 4 weeks + TCS, a weight-based regimen of dupilumab every 2 weeks + TCS, or placebo + TCS [31]	- For q4w and q2w there were higher IGA scores of 0 or 1 compared to placebo [31] - A higher percentage of q4w and q2w achieved ≥75% improvement in EASI and achieved at least a 4-point reduction in the Peak Pruritus [31,32] Numerical Rating Scale compared to placebo

IL—interleukin; R—randomized; PC—placebo controlled; DB—double blind; TCS—topical corticosteroids; IGA—investigator’s global assessment; EASI—eczema area and severity index score; q4w—patients dosed with 300 mg of dupilumab every four weeks with topical corticosteroids; q2w—patients dosed with a weight-based regimen of dupilumab every two weeks with topical corticosteroids.

**Table 2 biomedicines-09-00958-t002:** Biologics indicated for the treatment of psoriasis in children.

Drug Name	Biologic Agent Structure	Pediatric Indication	Study Type	Age of Participants	Length of Time	Methods	Results of Clinical Trials
TNF-a inhibitor							
Adalimumab	Human IgG1 monoclonal antibody [32]	This treatment is EMA approved for children 4 years of age and up, but not FDA approved. [33]	Phase III R	Children aged 4–18 years with severe plaque psoriasis [32]	-52 weeks -The initial 16 weeks of treatment was followed by a 36-week withdrawal period and a subsequent 16-week retreatment period [32]	Patients were randomized to an adalimumab 0.8 mg/kg or 0.4 mg/kg treatment every other week or to a methotrexate 0.1–0.4 mg/kg weekly [32]	PASI 75 from baseline was maintained or improved from the start to the end of the study [32]
TNF-a receptor fusion protein							
Etanercept	Fusion antibody combining human TNF-α receptor and Fc portion of monoclonal antibody [34]	FDA approved for the treatment of moderate to severe pediatric psoriasis for patients who are between 4–17 years of age [34]	Phase IIIR, P	Children aged 4–17 diagnosed with psoriasis [35]	48 weeks	- Patients were randomized to either 12 once-weekly SC injections of placebo or 0.8 mg of etanercept per kg of body weight, followed by 24 weeks of once-weekly open-label etanercept [35] - At week 36, 138 patients went through a second randomization to placebo or etanercept [35]	- 57% of patients receiving etanercept achieved a PASI 75 compared with only 11% receiving placeb [35] - By week 36 the rates of PASI 75 were 68% for the treated group and 65% for the placebo group [35]
IL-17A inhibitors							
Secukinumab	Fully human monoclonal antibody [36]	FDA approved for psoriasis in children six years of age and up [36]	Phase III R, PC	Children aged 6 to 18 years with severe chronic plaque psoriasis [36]	52 weeks	40 Patients were randomized to receive low-dose secukinumab, 40 were given high-dose secukinumab, and 41 were in the placebo or etanercept group [36].	- At week 12, the patients treated with secukinumab had a higher rate of PASI 75 than the patients treated with placebo [37] - At 52 weeks, the secukinumab group achieved higher rates of PASI 75, PASI 90, and PASI 100 [37].
Ixekizumab	High-affinity monoclonal antibody [38]	FDA approved for children 6 years and older with moderate to severe plaque psoriasis [39]	Phase III R, DB, P	Patients aged 6 to <18 years with moderate-to-severe plaque psoriasis [39]	12 weeks	Patients were randomized to weight-based dosing of IXE Q4W or placebo through week 12, followed by open-label IXE Q4W maintenance period [39]	- IXE was more effective than placebo for both PASI 75 and static Physician’s Global Assessment score (0,1) [39] - These improvements lasted for up to 48 [39]
IL-12 and IL-23 inhibitor							
Ustekinumab	Human monoclonal antibody that binds to and suppresses the p40 subunit [35]	FDA approved in patients 6 years and up for psoriasis [40]	Phase III CADMUS trial	Psoriatic adolescents aged 12–17 years [41]	12 weeks	Patients were placed in either SD or HSD groups [41]	By week 12, PASI 75 was achieved in more than 75% of patients in full and half dosing groups and PASI 90 was achieved in more than half of patients in full and half dosing groups [41]

TNF-a—tumor necrosis factor alpha; R—randomized; PC—placebo controlled; DB—double blind; PASI—psoriasis area and severity index; PASI 75—a 75% reduction in psoriasis area and severity index score; PASI 90—a 90% reduction in psoriasis area and severity index score; PASI 100—a 100% reduction in psoriasis area and severity index score; IL—interleukin; SC—subcutaneous; IXE—Ixekizumab; IXEQ4—Ixekizumab every 4 weeks; SD—standard dosing; HSD—half standard dosing.

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
