# Peer review of "Refractory Pediatric Psoriasis and Atopic Dermatitis: The Importance of Therapeutical Adherence and Biological Management"

_biomedicines, 2021, doi:10.3390/biomedicines9080958_

Round 1
Reviewer 1 Report
The article is not correctly formatted according to journal guidelines.
A small review talking about possible therapeutic treatments in AD and psoriasis in infants. I think the title of the article is measleading, as the article focuses mainly on the importance of therapeutical adherence, and then reports all biological options already available for psoriasis. I think that a title such as: " Refractory Pediatric Psoriasis and Atopic Dermatitis: the importance of therapeutical adherence and biological management", as basically no topical drugs, no traditional systemic drug were assessed for both diseases.
It would probably be better to include at least two or three search engines (not only Pubmed, but also Scopus, Google Scholar).
Page 2 line 48 : you should add: "topical steroids are usually associated with vitamin D derivatives in the management of psoriasis" and cite an article such as: doi: 10.1111/dth.13185.
Page 10 line 345-349 "Moderate to severe AD or psoriasis may be treated in a stepwise approach with systemic therapies, phototherapy, or biologics. Stronger biologic drugs indicated for treating pediatric psoriasis include adalimumab, secukinumab, etanercept, ustekinumab, and ixekizumab. These drugs provide specific inhibition of immune-mediated pathways involving tumor necrosis factor, IL-17, and IL-23." this paragraph needs a reference, such as: doi: 10.3390/healthcare9050543.
Thank You
Author Response
We have formatted the article according to the journal guidelines.
We have changed the title to " Refractory Pediatric Psoriasis and Atopic Dermatitis: the importance of therapeutical adherence and biological management" to convey the emphasis on therapeutic adherence in this review.
We have also included the search engine Scopus in the methods section.
We have added "topical steroids are usually associated with vitamin D derivatives in the management of psoriasis” and cited the article doi: 10.1111/dth.13185 on Page 2, line 48
We have added the reference, doi: 10.3390/healthcare9050543 to the paragraph on page 10, line 345-349
Thank You
Reviewer 2 Report
Thank you for your work.
Lines 309-312 is unclear. Please clarify if it is "initial clinical impressions" instead of "initial pathologic specimens". The biopsies already confirmed mycosis fungicides, whereas the initial impression was AD/psoriasis.
Line 150-154 please keep form of citation of references consistent.
Your discussion on the reasons for failure of topical therapies included "steroid phobia". You have not explored what patients and their parents/caregivers might do, when they have serious concerns regarding side effects of medications. This extends to other therapies discussed in your paper.
Complementary and alternative medicine (CAM) approaches were not discussed as potential causes of patient non-adherence (13% of outpatient dermatology patients reported using CAM) [Sivamani RK, Morley JE, Rehal B, Armstrong AW. Comparative Prevalence of Complementary and Alternative Medicine Use Among Outpatients in Dermatology and Primary Care Clinics. JAMA Dermatol.2014;150(12):1363–1365. doi:10.1001/jamadermatol.2014.2274]. Most frequently, herbal, homeopathic and ayurvedic therapies were the most commonly used, while eczema was the commonest reason for using CAM. About 42% of these patients in general use CAM without referring to their physician and many would prefer their physicians have knowledge of CAM and be open to discussing it with them [Nwabudike LC, Tatu AL. Using Complementary and Alternative Medicine for the Treatment of Psoriasis: A Step in the Right Direction. JAMA Dermatol. 2019;155(5):636. doi:10.1001/jamadermatol.2019.0106]
There is also work that suggests some of these therapies might have some effect on AD and psoriasis.
Thus, a discussion from the point of view of the patients and their caregivers would probably serve to complete the discussion and improve understanding of why patients may not be so adherent. The ball is not necessarily only the court of the patients. Perhaps physician lack of openness to other alternatives the patients may prefer may also partly be driving adherence.
Author Response
On Line 309-312, we have changed it from initial clinical impressions to initial pathologic specimens.
We have adjusted the form of citation on Line 150-154 to keep it consistent with the rest of the review
We have included an additional section of our discussion addressing responses to steroid phobia, which may include seeking alternative forms of medicine for both AD and psoriasis patients(cited doi:10.1001/jamadermatol.2014.2274 and doi:10.1001/jamadermatol.2019.0106).
Thank You